# Melatonin Promotes Antler Growth by Accelerating MT1-Mediated Mesenchymal Cell Differentiation and Inhibiting VEGF-Induced Degeneration of Chondrocytes

**DOI:** 10.3390/ijms23020759

**Published:** 2022-01-11

**Authors:** Xuyang Sun, Xiaoying Gu, Keke Li, Mengqi Li, Jingna Peng, Xinxin Zhang, Liguo Yang, Jiajun Xiong

**Affiliations:** Key Lab of Agricultural Animal Genetics, Breeding and Reproduction of Ministry of Education, College of Animal Science and Technology, Huazhong Agricultural University, Wuhan 430070, China; sunxuyangabc@163.com (X.S.); gxy19970310@163.com (X.G.); likekeasd@foxmail.com (K.L.); LX372328@163.com (M.L.); peng980906@163.com (J.P.); zxx1144795936@163.com (X.Z.)

**Keywords:** deer antler, melatonin, mesenchymal cell, vascular endothelial growth factor, chondrocytes

## Abstract

The sika deer is one type of seasonal breeding animal, and the growth of its antler is affected by light signals. Melatonin (MLT) is a neuroendocrine hormone synthesized by the pineal gland and plays an important role in controlling the circadian rhythm. Although the MLT/MT1 (melatonin 1A receptor) signal has been identified during antler development, its physiological function remains almost unknown. The role of MLT on antler growth in vivo and in vitro is discussed in this paper. In vivo, MLT implantation was found to significantly increase the weight of antlers. The relative growth rate of antlers showed a remarkable increased trend as well. In vitro, the experiment showed MLT accelerated antler mesenchymal cell differentiation. Further, results revealed that MLT regulated the expression of Collage type II (Col2a) through the MT1 binding mediated transcription of Yes-associated protein 1 (YAP1) in antler mesenchymal cells. In addition, treatment with vascular endothelial growth factor (VEGF) promoted chondrocytes degeneration by downregulating the expression of Col2a and Sox9 (SRY-Box Transcription Factor 9). MLT effectively inhibited VEGF-induced degeneration of antler chondrocytes by inhibiting the Signal transducers and activators of transcription 5/Interleukin-6 (STAT5/IL-6) pathway and activating the AKT/CREB (Cyclin AMP response-element binding protein) pathway dependent on Sox9 expression. Together, our results indicate that MLT plays a vital role in the development of antler cartilage.

## 1. Introduction

The deer antler is a bone organ propagating from a perpetual cranial bony bump named a pedicle [1]. One stunning feature of the deer antler is that it regenerates completely once lost every year [2]. Antler regeneration originates from the proliferation and differentiation of antler stem cells [3]. Recent findings have shown that antler stem cells could arise from cranial neural crest cells, multipotent to produce multiple cells, including neural cells, adipocytes and chondrocytes [4]. The tip of antler tissue is firstly stored by transforming antler stem cells into mesenchymal cells during antler genesis, which turns into pre-cartilage cells and finally transforms into chondrocytes [5]. Due to its quick regeneration in a short time, the deer antler can be used as a model to explore the development process of bone.

Melatonin is a neuroendocrine hormone synthesized by the pineal gland and conveys circadian rhythm information to the organism through external light signals maintaining control [6,7]. It also plays a crucial role in regulating the cardiovascular system [8], immunomodulation [9] and bone development [10]. In addition, it is involved in the biological function including antioxidant properties, anti-aging and anti-tumor effects [11,12,13]. Previous studies indicated that the changes of seasonal photoperiod stimulate the regeneration of deer antler, which is mainly related to the secretion of gonadotropins in the body [14]. Moreover, melatonin can act as a regulator to stimulate the secretion of gonadotropins and other hormones [15]. The association analyses between a mutation in the melatonin 1A receptor gene (MTNR1A or MT1) and sika velvet yield showed that the SNP (Single nucleotide polymorphism) of MTNR1A was related to its production. Therefore, MTNR1A could affect the growth of deer antler [14]. Despite all this, the role of melatonin in the development of deer antler is mainly unknown.

Compared with cartilage tissue of other species, there are many blood vessels in the cartilage tissue of deer antler, indicating that VEGF is an important growth factor for antler development. It is known that VEGF acts as a pro-angiogenic factor in angiogenesis and the proliferation of endothelial cells [16]. In addition, VEGF plays a vital role in regulating the cartilage development [17]. Studies have shown that bone formation is impaired by inhibiting VEGF activity, and the expansion of hypertrophic chondrocytes is completely inhibited [18]. Moreover, VEGF can also promote the survival of chondrocytes by inhibiting apoptosis [19]. Recent studies have shown that MLT regulates VEGF expression and mediates angiogenesis [20,21]. Furthermore, MLT could improve VEGF-induced abnormal vascular permeability [22]. Nevertheless, whether they will cooperate in regulating the cartilage development of antler is still unclear.

In this study, we aimed to investigate the physiological function of MLT during antler development. We demonstrate that MLT accelerates the differentiation of antler mesenchymal cells into chondrocytes. Strikingly, during late chondrogenesis, VEGF induces chondrocytes degeneration by downregulating the expression of Col2a and Sox9. MLT treatment effectively relieves symptoms and maintains chondrocyte phenotype. Furthermore, we assessed the effects of MLT on VEGF-induced chondrocytes degeneration.

## 2. Results

### 2.1. Subcutaneous Implantation of Melatonin and Endocrinological Profiles

In this study, 21 male sika deer were randomly divided into three groups. Different doses of melatonin sustained-release agents (0 mg, 40 mg and 80 mg) were implanted subcutaneously in the neck of sika deer. Subsequently, the secretion levels of melatonin and testosterone in the plasma were measured on days 0, 10 and 60. On days 10 and 60 days, the melatonin level of Group 3 was significantly higher than that of other groups (Figure 1A). However, on day 10, the testosterone level of Group 3 was significantly lower than that of Group 1 and 2. On the 60th day, the testosterone level of Groups 2 and 3 was significantly lower than that of Group 1 (Figure 1B). After implantation, exogenous melatonin significantly reduced testosterone concentration in the plasma during the velvet stage, which also indicated that low testosterone concentration was beneficial to antler growth.

### 2.2. Effect of Melatonin Implantation on Antler Growth

The results of the current study showed that melatonin implantation could significantly affect the growth of deer antler. After 60 days of melatonin treatment, the antler were sawed off and weighed. The antler weight of Groups 2 and 3 increased significantly compared with that of Group 1 (*p* < 0.05). There was a significant increase in relative antler weight (*p* < 0.05) as well. Although there was no statistical difference between Groups 2 and 3 implanted with melatonin, there was a dose-dependent relationship (Table 1). The length of the antler was measured on day 20 after melatonin implantation. Subsequently, the measurement was repeated at the interval of five days to calculate the relative growth rate of the antler. Although there were no statistically significant changes in the relative growth rate of the antler, the increased trend in Group 2 and 3 was more evident than that of Group 1 (Table 2).

### 2.3. MLT Accelerates Antler Mesenchymal Cells Differentiation In Vitro

The antler cells obtained by layered culture, including mesenchymal cells, pre-chondrocytes and chondrocytes, were stained with toluidine blue and alcian blue. There was no purple–red phenomenon in the cytoplasm of mesenchymal cells after toluidine blue staining compared with pre-chondrocytes and chondrocytes. After staining with alcian blue, the mesenchymal cells showed no positive reaction compared with pre-chondrocytes and chondrocytes (Figure 2A). We evaluated the expression level of Col2a (chondrocyte marker molecule) in different tissue layers using Western blot assay, confirming that the rapid growth of deer antler is accompanied by the differentiation of mesenchymal cells into chondrocytes (Figure 2B). For exploring the MLT role in the growth of antlers, antler mesenchymal cells were cultured in vitro, with or without MLT (1 μM) for 3 days. MLT treatment increased the expression of Col2a protein compared with the negative control (NC) in mesenchymal cells (Figure 2C). Furthermore, results of toluidine blue staining showed that purple–red discoloration appeared in the cytoplasm, and alcian blue staining had positive reaction after MLT treatment (Figure 2D). This indicates that MLT accelerates the differentiation of antler mesenchymal cells.

### 2.4. MLT Regulates the Expression of Col2a through the MT1 Binding Mediated YAP1 Pathway in Antler Mesenchymal Cells

To determine whether MLT regulates the expression of Col2a through MT1, antler mesenchymal cells were transfected with MT1 interference fragment. The results of Western blot and immunofluorescence staining indicated that MT1 was effectively knocked down after 72 h of transfection (Figure 3A,B). The Western blot assay showed that MLT treatment significantly increased the expression of Col2a, whereas MT1 knockdown markedly decreased Col2a expression, even if MLT was added exogenously (Figure 3C). These results indicate that MLT upregulates the expression of Col2a through the MT1 binding.

Next, we investigated how the MLT/MT1 complex transmits signals to the nucleus. Recent studies have indicated that YAP1 protein is a powerful signaling intermediate, which can enter the nucleus from the cytoplasm and play important roles in regulating cell differentiation, organ growth and stem cell self-renewal [23,24]. The results showed that MLT treatment increased the expression of YAP1, while MT1 knockdown inhibited melatonin-mediated YAP1 expression (Figure 3D). To further confirm the role of YAP1 in antler mesenchymal cells, we used RNAi to knock down the expression of YAP1. Results of Western blot showed YAP1 knockdown reduced the Col2a expression (Figure 3E). These results indicate that MLT regulates the expression of Col2a through the MT1 binding mediated YAP1 pathway in the mesenchymal cells of antler.

### 2.5. MLT Inhibits VEGF-Induced Degeneration of Antler Chondrocytes

Immunohistochemistry results showed that many blood vessels were distributed in the tissue of deer antler, and the vascular cavity in the cartilage layer is thicker (Figure 4A). Furthermore, the expression of VEGF in antler chondrocytes was significantly increased compared with mesenchymal cells analyzed by qPCR (Figure 4B). For determining the role of VEGF in the development of antler cartilage, a CCK-8 assay was used to detect cell viability of antler chondrocytes after being treated with VEGF165 by different concentrations for 24 h. The results showed that the cell viability in the 5 ng/mL VEGF groups was considerably increased compared to that in the control group (Figure 4C). Under in vitro conditions, the treatment with exogenous VEGF treatment promoted chondrocytes contraction and changed the morphology of antler chondrocytes; toluidine blue staining result showed that the morphology of chondrocytes changed from a long spindle to an irregular round (Figure 4D,E). The expressions of Col2a and Sox9 were significantly decreased after treatment with VEGF, which revealed that VEGF induces degeneration of antler chondrocytes (Figure 4F). Subsequently, we examined whether MLT affected VEGF-induced degeneration of antler chondrocytes. The Western blot result confirmed that MLT inhibited VEGF-induced degeneration of antler chondrocytes, and MLT did not affect the expression of VEGF in the antler chondrocytes (Figure 4G,H).

### 2.6. Effects of MLT on VEGF-Induced Chondrocytes Degeneration through Inhibition of p-STAT5/IL-6 and Activation of p-AKT/p-CREB Dependent on Sox9 Expression

We studied the regulatory mechanism that MLT inhibits VEGF-induced degeneration of antler chondrocytes. Oxidative stress is a common factor that induces cartilage degradation. Reactive oxygen species (ROS) assay was performed with or without VEGF treatment and observed that VEGF-induced ROS production (Figure 5A). Furthermore, we examined the expression of peroxiredoxins (PRDXs) after VEGF treatment by q-PCR. We found that VEGF treatment decreased the expression of PRDX2 and PRDX4. The downregulation of PRDX2 and PRDX4 resulted in the accumulation of ROS. In comparison, VEGF combined with MLT reduced the ROS level by upregulating the expression of PRDX2 and PRDX4 (Figure 5A,B). In addition, MLT significantly inhibited VEGF-regulated interleukin6 (IL-6) expression (Figure 5C). Although VEGF treatment inhibited Sox9 expression, VEGF combined with MLT significantly increased Sox9 expression (Figure 5D). STAT pathway mediates the expression of multiple inflammatory factors, VEGF induces IL-6 expression possibly by activating the STAT pathway. As shown in Figure 5E, MLT significantly inhibited VEGF-induced activation of p-STAT5 in antler chondrocytes. It is known that cAMP response element binding protein (CREB) binds to the promoter site upstream of Sox9 and regulates Sox9 expression [25]. AKT/CREB pathways are probably involved in VEGF-inhibited Sox9 expression. Western blot results showed that MLT significantly increased the VEGF- induced inactivation of AKT/CREB in antler chondrocytes (Figure 5E). Our results verified that MLT inhibits VEGF-induced chondrocytes degeneration through inhibition of p-STAT5/IL-6 and activation of p-AKT/p-CREB dependent on Sox9 expression.

## 3. Discussion

Recent studies have accentuated the importance of MLT in multiple biological processes, specifically in regulating circadian rhythms, hormone endocrine, neuroimmunological and reproductive processes [26,27,28,29]. Additionally, MLT can modulate cell proliferation, differentiation and apoptosis by sequential activation of multiple signaling molecules [30,31]. These authors mainly focused on cancer treatment and disease defense, but little is known about antler development. The photoperiod is an important factor affecting antler growth. Yang et al. found that the SNP of MTNR1A was related to antler production through the association analysis and believed that MTNR1A might affect growth of deer antler [14]. In this study, we subcutaneously implanted different doses of melatonin sustained-release agents in the neck of sika deer. We found that MLT treatment significantly increased the weight of antler, and the relative growth rate of antler also had a remarkable trend of increase (Table 1 and Table 2). Based on our results, MLT was identified as a promoter of antler development.

Antler mesenchymal cells are located at the tip of antler tissue and can quickly differentiate into chondrocytes, fibroblasts and osteocytes, thereby considered to be the source of antler tissue [32,33]. MLT has been proven to regulate mesenchymal stem cell differentiation [34]; however, its regulatory mechanism on cell differentiation was not uniform. Dong et al. found that MLT could promote osteoblastic differentiation of rat mesenchymal stem cells by upregulating the NPY (neuropeptide Y)/NPY receptor Y1 [35]. Moreover, Zhu et al. found that MLT mediated osteoblastic differentiation of MC3T3-E1 cells by activating PDGF (Platelet-derived growth factor)/AKT signaling pathway [36]. In the previous study, we identified the effective expression of the MT1 receptor in the growth process of antler [5]. Herein, MT1 knockdown blocked the expression of Col2a regulated by MLT, indicating that MLT promoted the differentiation of antler mesenchymal cells into chondrocytes depending on the receptor (Figure 3C). YAP/TAZ signaling plays a crucial role in cell differentiation. Results of related research showed that YAP/TAZ proteins could be used as a signal intermediate to link adhesion and mechanical cues to the differentiation of mesenchymal stem cells [37,38]. Ma et al. found that Luzindole (melatonin receptor inhibitor) treatment could significantly block the transcriptional activity of YAP1 [39]. Therefore, YAP1 may serve as a downstream target protein of MLT/MT1 signal pathway to transmit information.

As shown in Figure 4A, antler growth is accompanied by the invasion of blood vessels. These blood vessels deliver plenty of nutrients for the growth of deer antler, which makes antler stem cells capable of generating approximately 10 kg of antler tissue within 60 days [40]. VEGF as a pro-angiogenic factor is involved in angiogenesis and the formation of blood vessels [16]. In addition, VEGF is involved in cartilage development and bone formation [18]. In this study, we found that VEGF(5 ng/mL) treatment increased the level of ROS (Figure 5C). In vivo and vitro studies showed that oxidative stress can damage the normal development of cartilage and cause osteoarthritis by inducing an inflammatory response [41,42]. Moreover, oxidative stress products also promote the degeneration of chondrocytes and induce apoptosis [43,44]. These results indicate that oxidative stress is a common pathological case that damages cartilage development, including antler cartilage. Therefore, we tested the function of MLT on the growth of chondrocytes in a VEGF-induced environment of oxidative stress.

The release of inflammatory cytokines contributes to the dedifferentiation of chondrocytes and degradation of the extracellular matrix [45]. In this article, we found that VEGF treatment promoted the expression of the inflammatory factor IL-6 (Figure 5B). We speculated that the up-regulation of IL-6 was related to oxidative stress and the STAT5 pathway. STATs are a family of nuclear transcription proteins that mediate the expression of multiple cytokines, including ILs and IFNs [46]. Previous studies have shown that the STAT5 pathway regulates the expression of IL-6. For example, LPS induces IL-6 expression by activating JAK2/STAT5 pathway, and Caveolin-1 deficiency reduces the resistance to K. pneumoniae infection by activating the STAT5 pathway to produce pro-inflammatory cytokines [47,48]. Furthermore, Krylatov et al. proposed that ROS-induced by cytokines or growth factors plays an important signaling role in regulating JAK (Janus Kinases), STAT3 and STAT5 biological activity [49]. We found that MLT inhibited VEGF-induced IL-6 expression by inhibiting ROS production and the STAT5 pathway.

Sox9 is a critical transcription factor in endochondral ossification. It is the basis for the growth and differentiation of chondrocytes through transcriptional activation of several genes necessary for maintaining healthy cartilage [50]. Sox9 can bind the promoter region of Col2a to regulate the expression of Col2a and secures chondrocyte lineage commitment [51]. The cAMP/CREB pathway is a crucial regulator of cell proliferation, migration and survival. Previous studies have shown that sequence analysis of the proximal promoter regions of mouse Sox9 and human Sox9 revealed that this sequence has a binding site for CREB [52]. CREB inhibition could down-regulate the transcriptional activity of Sox9 and inhibit cartilage development. The PI3K/AKT pathway is a common regulator that mediates the activation of CREB. Xue et al. found that Lactoferrin relieves Il-1β regulated chondrocytes apoptosis by activating AKT-induced CREB [53]. These studies are similar to our results, confirming that MLT inhibits VEGF-induced chondrocytes degeneration by activating the AKT/CREB/Sox9 pathway.

In conclusion, we have identified that MLT promotes antler growth through subcutaneous implantation of melatonin. In vitro experiment, our results indicate that MLT plays two roles in regulating antler growth (Figure 6). On the one hand, MLT accelerates antler mesenchymal cell differentiation by regulating Col2a expression, promoting cartilage development. On the other hand, MLT inhibits VEGF-induced chondrocytes degeneration, ensuring the process of endochondral ossification.

## 4. Materials and Methods

### 4.1. Subcutaneous Implantation of Melatonin

Twenty-one healthy, 2 to 3 year old male sika deer (with similar antler production and body weight) were selected as the experimental subjects, provided by the sika deer breeding farm of Wuhan Jinsanxin. In this experiment, the 21 male sika deer were randomly divided into 3 groups (Group 1, Group 2 and Group 3). According to the experimental design, when the sika deer did not fall off the plate in mid-March, the three groups of experimental deer were given melatonin under the neck skin. Different doses of melatonin sustained-release agents (Institute of special animal and plant science, Chinese Academy of Agricultural Sciences, Changchun, China) were implanted in the implants. The grouping condition is shown in Table 3.

### 4.2. Detection of Hormone Level in Plasma

The levels of melatonin and testosterone in plasma were detected by ELISA method, according to the kit operation method provided by the manufacturer (Enzyme Link Biotechnology, Shanghai, China).

### 4.3. Cell Separation and Culture

The tip of antler tissues growing about 60 days was dissected in different layers (mesenchyme layer, pre-cartilage, and cartilage layer) separately using the treatment described previously [54]. Dissected tissues were cut up in DMEM/high glucose medium (Hyclone, GE Healthcare, Logan, UT, USA) and digested by 0.2% collagenase II (Sigma–Aldrich, Marlborough, MA, USA) at 37 °C; for 30 min. Digested cells were collected and resuspended in a medium containing 10% FBS (fetal bovine serum). Antler cells were cultured at 37 °C with 5% CO_2_.

### 4.4. Immunohistochemistry Staining

First, paraffin sections were deparaffinized, and the sections were exposed to 3% hydrogen peroxide and washed three times with PBS for 5 min each. Paraffin sections were blocked with PBS containing goat serum for 40 min, and incubated with CD31 antibody (Service bio, Wuhan, China) at 4 °C overnight. Subsequently, the paraffin sections were washed 3 times with PBS, goat anti-rabbit IgG was added and incubated at 37 °C for 30 min, then rinsed with PBS. Before neutral staining, paraffin sections were counter-stained with hematoxylin for 2 min, washed with ddH2O and differentiated with 1% hydrochloric acid alcohol for 8 s, washed with ddH2O, returned to blue with ammonia and finally washed with ddH2O.

### 4.5. RNA Interference

Cells were seeded in cell culture plates until 60–70% confluence. The cells were transfected with MT1 or YAP1 siRNA by RNAiMAX Reagent in an Opti-MEM medium (Life Technologies, Inc., Carlsbad, CA, USA) according to the instructions. After 72 h of transfection, cells were harvested for mRNA expression or protein expression. The siRNA target sequences of MT1 and YAP1 were synthesized (GenePharma Co., Ltd., Shanghai, China). MT1: 5′-CCUCAAUGCGAUCAUAUAUTT-3′, YAP1: 5′-GGUGACACUAUCAACCAAATT-3′.

### 4.6. Cell Viability Test

Cells were seeded in 96-well plates (10,000/well). After treatment with VEGF (BBI Life Sciences, Shanghai, China) for 24 h, cell viability was tested by cell counting kit-8 (CCK-8) (Dojindo, Kyushu, Japan). The viable cells were determined using standard enzyme instruments to detect the λ = 450 nm absorbance values.

### 4.7. Total RNA and Quantitative Real-Time PCR (q-PCR)

Total RNA was extracted by RNA extraction kit (Cat R6834-02, Omega Bio-Tek, Norcross, GA, USA) according to the kit instructions. Subsequently, the purity of RNA was tested the absorbance at 260/280. The standard RNA was used to reverse transcribed into cDNA by cDNA first-trans synthesis kit (CatKR118-02, TIANGEN Biotech, Co., Ltd., Beijing, China). The primers used in this study are given in Table 4.

### 4.8. Western Blot Assay

Pretreatment of antler chondrocytes was carried out with MLT (Meilunbio, Dalian, China) for 2 h, then VEGF was added for 24 h. The treated cells were exposed to lysis buffer (RIPA lysis and 50 × Cocktail) (Servicebio, Wuhan, China) for 20 min. Extract protein and detect protein concentration by the BCA protein assay (Servicebio, Wuhan, China). The total protein (20 μg) was separated by SDS-PAGE. Subsequently, the separated proteins were transferred to polyvinylidene difluoride (PVDF) membranes and blocked with 5% skimmed milk powder in TBS for 2 h and incubated PVDF membrane with primary antibody at 4 °C overnight. The following antibodies were used: anti-Melatonin Receptor 1A (1:1000, GTX31054, GeneTex, San Antonin, TX, USA), anti-Col2a (1:2000, 15943, Proteintech, Wuhan, China), anti-YAP1 (1:1000, GTX129151, GeneTex), anti-p-AKT (Ser473) (1:1000, 4070, Cell Signaling Technology (CST) Biological reagents Company Ltd., Shanghai, China), anti-AKT (1:1000, C67E7, CST), anti-p-CREB(Ser133) (1:1000, 9198, CST) and anti-p-STAT5(Tyr694) (1:1000, 4322, CST), anti-STAT5 (1:1000, 25656, CST). After washing for three times by TBST, membranes were incubated with HRP-conjugated secondary antibodies for 2 h. At last, the membranes were treated with ECL chemiluminescence reagent and exposed to X-ray film to observe protein bands.

### 4.9. Reactive Oxygen Species

Reactive oxygen species (ROS) were detected using DCFH-DA (S0033S, Beyotime Biotechnology, Hefei, China). The DCFH-DA was diluted with serum-free culture medium to a final concentration of 10 μM. The cells were incubated with DCFH-DA dilution and incubated for 20 min at the 37 °C in the cell incubator. Subsequently, cells were cultured with Hoechst 33258 regent (Servicebio, Wuhan, China) for 10 min at the 37 °C in the cell incubator. Finally, cells were washed with serum-free cell culture solution 3 times before detecting ROS.

### 4.10. Immunofluorescence Assay

Cells were seeded on circular glass coverslips in 24-well plates. Cells were fixed with 4% paraformaldehyde for 15 min. After washing by PBS, permeabilized using TritonX-100 for 3 min. Subsequently the cell coverslips were blocked by 5% BSA for 30 min and then incubated with primary antibodies at 4 °C overnight. Cell coverslips were washed three times with PBST and incubated with CY3-conjugated secondary antibodies (Servicebio, Wuhan, China) for 2 h. The nucleus was labelled by DAPI (Servicebio, Wuhan, China). At last, the fluorescence signal was observed by laser scanning confocal microscope.

### 4.11. Statistical Analysis

Analysis of data was done using GraphPad Prism 5.0 software (GraphPad Software, Inc., San Diego, CA, USA). Data were summarized as mean ± standard deviation (SD), and a significant (*p* < 0.05) difference between the groups was determined following a one-way analysis of variation (ANOVA).

## Figures and Tables

**Figure 1 ijms-23-00759-f001:**
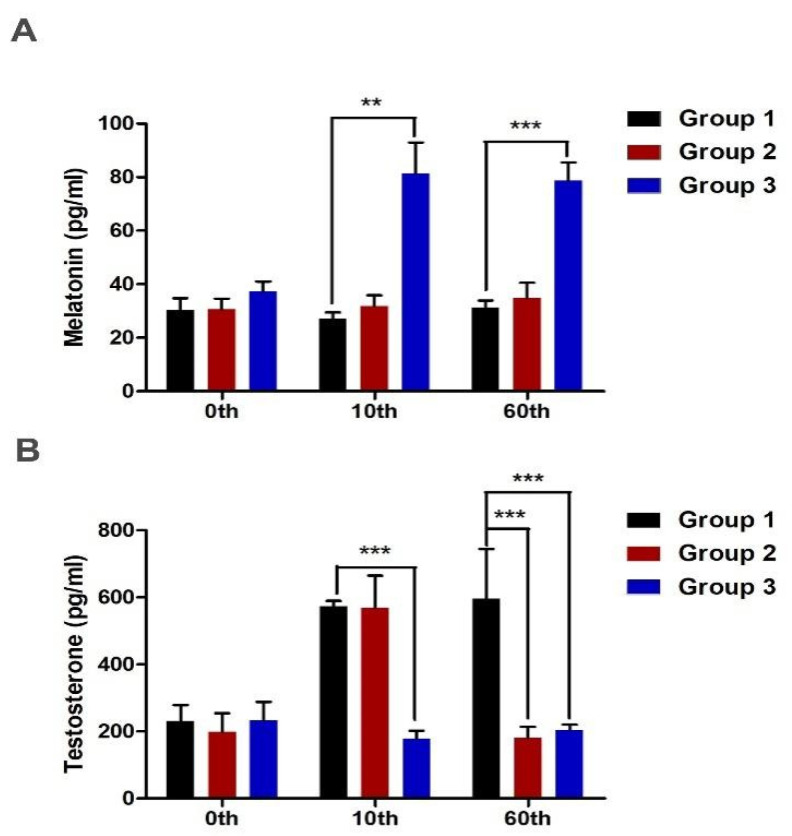
Subcutaneous implantation of melatonin and endocrinological profiles. (**A**) After implanting different doses of melatonin sustained-release agent, the concentration of melatonin in plasma was measured on day 0, day 10 and day 60. (**B**) After implanting different doses of melatonin sustained-release agent, testosterone concentration in plasma was measured on days 0, 10 and 60. N = 7 sika deer for test group, ** *p* < 0.01, *** *p* < 0.001.

**Figure 2 ijms-23-00759-f002:**
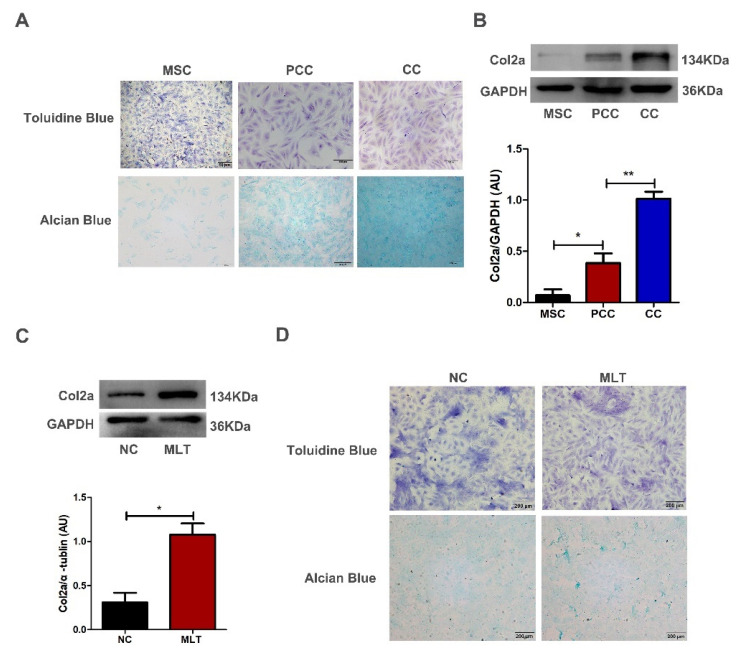
Melatonin accelerates antler mesenchymal cell differentiation in vitro. (**A**) MSC (mesenchymal cells), PCC (pre-chondrocytes) and CC (chondrocytes) were stained with toluidine blue and alcian blue; (**B**) Western blot was performed to examine the Col2a protein levels in MSC, PCC and CC, a-tubulin was used as an internal control. (**C**) The Col2a expression level was increased after MLT treatment in the MSC for 72 h by Western blot. (**D**) Toluidine blue and alcian blue staining of cells. Antler mesenchymal cells were treated with MLT (7 days), scale bar, 200 μm. The data include the mean ± SD of three independent experiments. * *p* < 0.05, ** *p* < 0.01.

**Figure 3 ijms-23-00759-f003:**
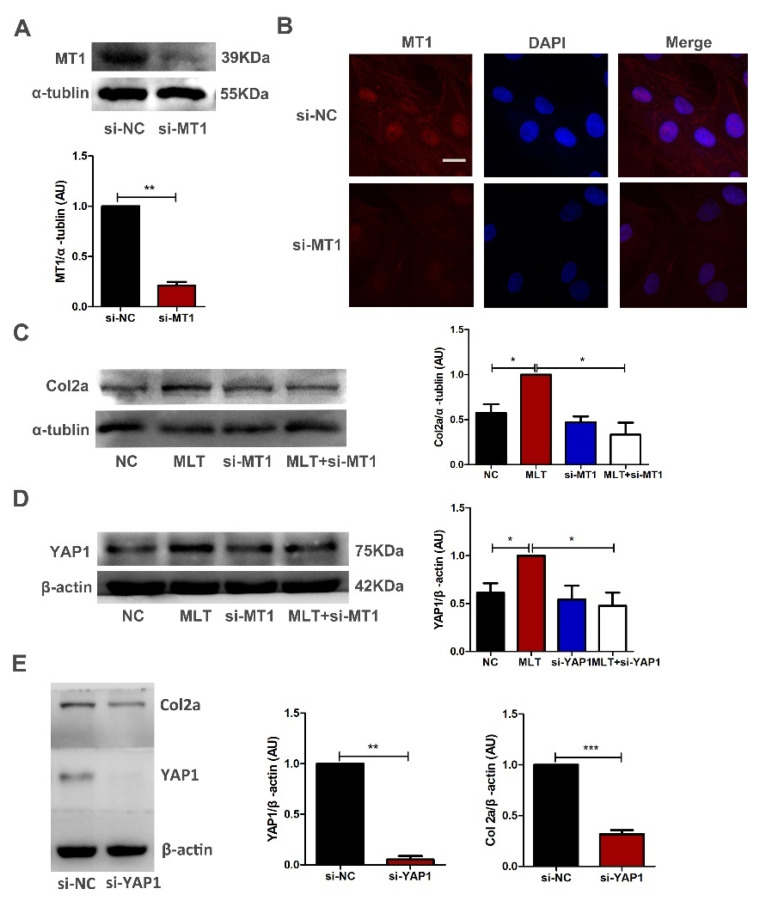
MLT regulates the expression of Col2a through the MT1 binding mediated YAP1 pathway in antler mesenchymal cells. (**A**) Western blot showed that MT1 expression level was significantly reduced after treatment si-MT1 in antler mesenchymal cell for 72 h. (**B**) MT1 siRNA was transfected for 72 h in antler mesenchymal cells. Localization of MT1 was determined by Immunofluorescence staining. The nuclear were stained with DAPI (blue), and the MT1 was detected with CY3 (red); scale bar, 20 μm. (**C**) Antler mesenchymal cells were transfected with MT1 siRNA or control, 24 h after transfection, cells were treated with MLT for 48 h. 3d after transfection, Col2a expression levels were analyzed by Western blot. (**D**) Antler mesenchymal cells were transfected with MT1 siRNA or control, 24 h after transfection, cells were treated with MLT for 48 h. 3d after transfection, YAP1 expression levels were analyzed by Western blot. (**E**) Antler mesenchymal cells were transfected with YAP1 siRNA or control, 72 h after transfection, YAP1 and Col2a expression levels were analyzed by Western blot. The data include the mean ± SD of three independent experiments. * *p* < 0.05, ** *p* < 0.01, *** *p* < 0.001.

**Figure 4 ijms-23-00759-f004:**
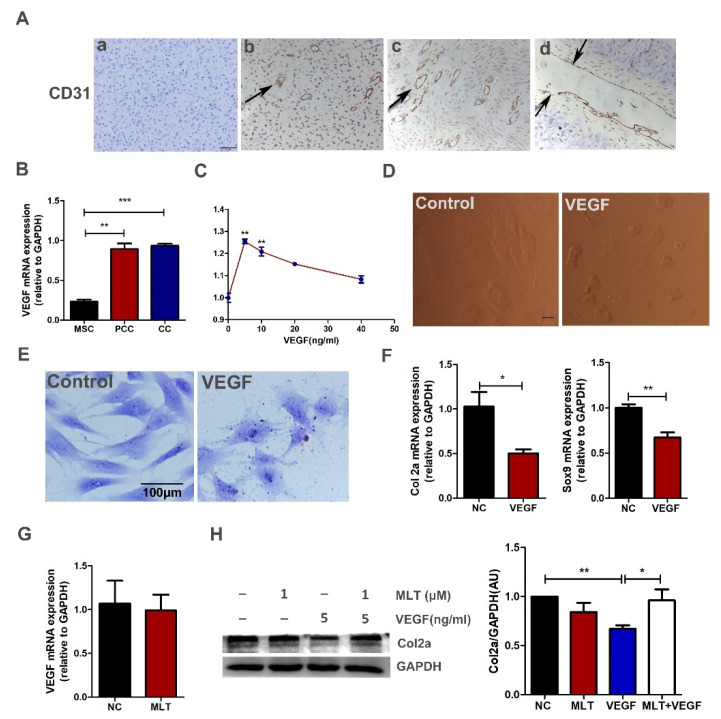
MLT inhibits VEGF-induced degeneration of antler chondrocytes. (**A**) Immunohistochemically expressed CD31 in all stages of velvet antler growth, a—representing negative control; b—representing mesenchymal layer; c—representing pre-chondrocyte layer; d—representing chondrocyte layer, scale bar = 50 μm. (**B**) qPCR of VEGF expression levels in mesenchymal cells, pre-cartilage cells and chondrocytes. (**C**) The CCK-8 assay was used to detect cell viability after VEGF165 treatment in different concentrations for 24 h. (**D**) Pictures of the morphology of antler chondrocytes treated with VEGF165 (5 ng/mL) through an inverted microscope, scale bar = 500 μm. (**E**) Toluidine blue staining of chondrocytes. Antler chondrocytes were treated with VEGF for 24 h, scale bar, 100 μm. (**F**) q-PCR was performed to examine the expression levels of Col2a and Sox9 after treatment with VEGF165 for 24 h in the chondrocytes. (**G**) q-PCR was performed to examine the expression level of VEGF after treatment with MLT for 24 h. (**H**) Antler chondrocytes were treated with MLT (1 μM), after 2 h incubation, cells were treated with VEGF (5 ng/mL) for 24 h; 1 day after treatment, Col2a expression levels were analyzed by Western blot. The data include the mean ± SD of three independent experiments. * *p* < 0.05, ** *p* < 0.01, *** *p* < 0.001.

**Figure 5 ijms-23-00759-f005:**
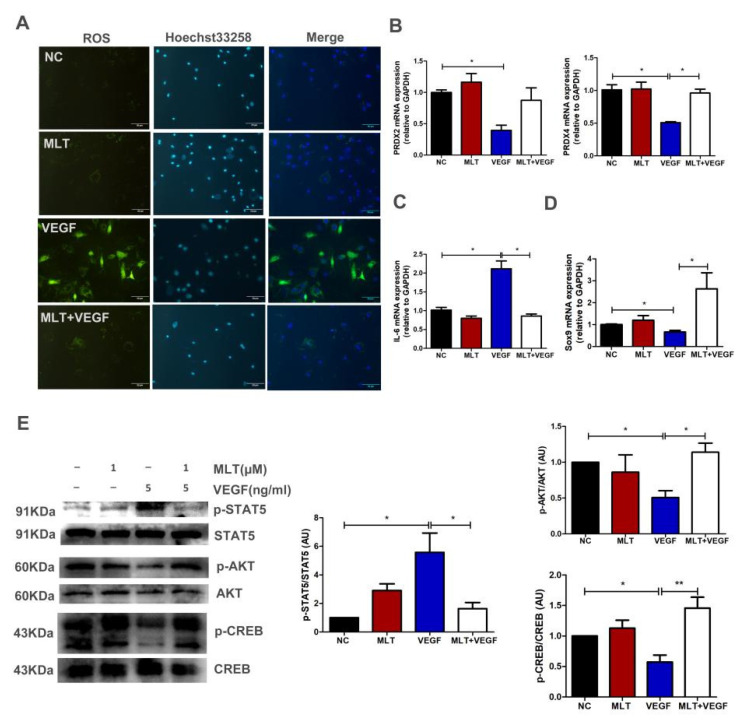
Effects of MLT on VEGF-induced chondrocytes degeneration through inhibition of p-STAT5/IL-6 and activation of p-AKT/p-CREB dependent on Sox9 expression. (**A**) Pretreatment of antler chondrocytes was carried out with MLT (1 μM) for 2 h, then VEGF (5 ng/mL) was added for 24 h. The level of reactive oxygen species was detected using ROS Assay Kit. Hoechst 33258 was used as a nuclear dye; scale bar, 50 μm. (**B**) PRDX2 and PRDX4 expression levels were analyzed by qPCR. (**C**) IL-6 expression level was analyzed by qPCR. (**D**) Sox9 expression level was analyzed by qPCR. (**E**) Pretreatment of antler chondrocytes was carried out with MLT (1 μM) for 2 h, then VEGF (5 ng/mL) was added for 24 h. The expression of p-AKT, p-STAT5 and p-CREB were detected via Western blot. The expression levels of AKT, STAT5 and CREB were used as loading controls, respectively. The data include the mean ± SD of three independent experiments. * *p* < 0.05, ** *p* < 0.01.

**Figure 6 ijms-23-00759-f006:**
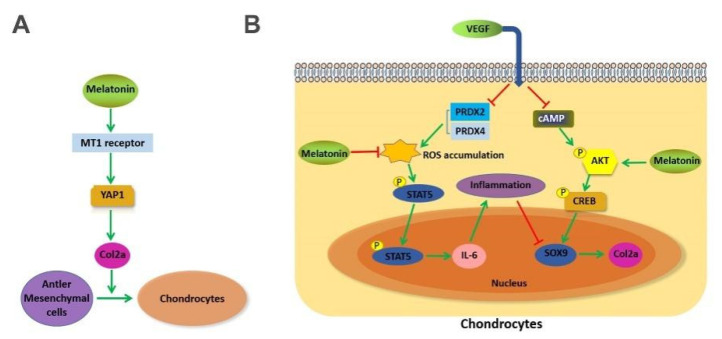
(**A**) MLT regulated the expression of Col2a through the MT1 binding mediated transcription of YAP1 in antler mesenchymal cells. (**B**) MLT effectively inhibited VEGF-induced degeneration of antler chondrocytes through inhibition of p-STAT5/IL-6 and activation of p-AKT/p-CREB dependent on Sox9 expression.

**Table 1 ijms-23-00759-t001:** The weight, relative weight and weight gain rate of antlers with different dosages of melatonin sustained-release agents (X ± S, *n* = 7).

Group	Number	Antler Weight (Kg)	Relative Weight	Weight Gain
Group 1 (0 mg/per)	7	1.13 ± 0.05 ^b^	1 ± 0.04 ^b^	-
Group 2 (40 mg/per)	7	1.68 ± 0.19 ^a^	1.49 ± 0.16 ^a^	49%
Group 3 (80 mg/per)	7	1.79 ± 0.51 ^a^	1.59 ± 0.45 ^a^	59%

Group 1—No melatonin slow-release agent implanted in each male deer. Group 2, 40 mg of melatonin slow-release agent implanted in each male deer. Group 3, 80 mg of melatonin slow-release agent implanted in each male deer. ^a, b^—indicating that there were significant differences within the column of different lowercase superscripts (*p* < 0.05).

**Table 2 ijms-23-00759-t002:** The relative growth rate of antlers with different doses of melatonin sustained- release agent (X ± S, *n* = 7).

Number	Group 1 (0 mg/per)	Group 2 (40 mg/per)	Group 3 (80 mg/per)
1	0.54 ± 0.05	0.56 ± 0.05	0.58 ± 0.01
2	0.23 ± 0.05	0.26 ± 0.05	0.39 ± 0.11
3	0.32 ± 0.11	0.37 ± 0.07	0.36 ± 0.16
4	0.16 ± 0.03	0.23 ± 0.15	0.27 ± 0.09
5	0.15 ± 0.05	0.16 ± 0.07	0.22 ± 0.09
6	0.07 ± 0.03	0.1 ± 0.05	0.13 ± 0.08
7	0.15 ± 0.06	0.13 ± 0.04	0.18 ± 0.07

Column header: 1–7 respectively indicated the number of measurements in the relative growth rate of antlers. After the melatonin slow-release agent was implanted, the antler length was measured every five days for eight times, the relative growth rate of seven time periods was calculated from this, and then arranged in chronological order. The relative growth rate was measured using the following method: W_n_ = (L_(n+1__)_ − L_n_)/L_n,_ W_n_ is the relative growth rate, L_n_ is the antler length at the n time measurement.

**Table 3 ijms-23-00759-t003:** Parameters of embedded melatonin sustained-release agent in each group.

Parameters	Group 1	Group 2	Group 3
Number	7	7	7
Melatonin dosage (mg/per)	0	40	80

**Table 4 ijms-23-00759-t004:** Primers used for RT-PCR in this study.

Gene	Primer Sequence (5′-3′)	Product Size
PRDX2	F: ACATTCCCCTGCTGGCTGAT	267 bp
	R: CGTCCACATTGGGCTTGATT	
PRDX4	F: TGATTCACAGTTCACCCATTTG	214 bp
	R:CACGGGAAGGTCATTCAGAGTA	
IL6	F: GCATTCCCTCCTCTGGTCA	228 bp
	R: AAAACATTCAAGCCGCACA	
Col2a	F: GAGGCAGCCGGCAACCTGAG	118 bp
	R: TGCGAGCTGGGTTCTTGCGG	
Sox-9	F: ACGCAGATTCCCAAGACACTA	143 bp
	R: ACGCTCGCTTTGAAGGTTT	
VEGF-A	F: AATGACGAAAGTCTGGAGTG	120 bp
	R: TTTGTTATGCTGTAGGAAGC	
GAPDH	F: GAAGGGTGGCGCCAAGAGGG	142 bp
	R: GGGGGCCAAGCAGTTGGTGG	

## Data Availability

Data will be available upon request to the corresponding author.

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
