# Peer review of "Melatonin Promotes Antler Growth by Accelerating MT1-Mediated Mesenchymal Cell Differentiation and Inhibiting VEGF-Induced Degeneration of Chondrocytes"

_ijms, 2022, doi:10.3390/ijms23020759_

Round 1
Reviewer 1 Report
In this work, the authors studied the effect of melatonin, a neuroendocrine hormone synthesized by the pineal gland, on antler growth, on differentiation of mesenchymal cells into chondrocytes and in the same time on the process of chondrocytes degeneration. The main findings are supported by the methodology. Overall the experiments are sound and well thought out but I have a number of small concerns with the manuscript in its current form.
Specific comments for revision:
- Authors should add molecular weight near protein bands on western blot membrane.
- Figure 1 - authors should delete from figure legend “**P<0.01, ***P<0.001” because of I don’t see on the figure any ** or ***
- On the figure 3B, Scale bar should be presented
- Authors should change the resolution of Figure 4D because of it is impossible to analyse the morphology of antler chondrocytes with these microphotographs
- Why authors did not use any nuclear living-cell stayning on Figures 5?
- To improve the representation of the experimental results with si-MT1 (Figure 3B), the authors should analyse the colocalisation coefficient between MT1 and DAPI stayning.
- The authors should edit the alphabetical symbols on the figure 5 and legend.
- Figure 5 G – pCREB should be normalised with total CREB
- In supplementary original-images of western blot:
- 2C, 3A, 3C, 3D – should be presented full membrane with marker.
- 3C – I don’t see difference between exposure 8 or 30 sec, even more the membrane after 8 sec represent more intensive bands than 30 sec
- Membrane under “Supplementary strips” “exposure 10s” – upper strip – which Ab autors used?
- Why in “Original-images pdf file” for western blotting results for figure 4G and 5G the same membrane is represented?
Author Response
Thank you for your comments concerning our manuscript entitled:Melatonin promotes antler growth by accelerating MT1-mediated mesenchymal cell differentiation and inhibiting VEGF-induced degeneration of chondrocytes. Your comments are all valuable and very helpful for revising and improving our paper, as well as the important guiding significance to our research. For your comment, we have also made a detailed modification.
- Authors should add molecular weight near protein bands on western blot membrane.
We have added protein molecular weight in the revised manuscript
- Figure 1 - authors should delete from figure legend “**P<0.01, ***P<0.001” because of I don’t see on the figure any ** or ***
We have made changes in the revised manuscript.
- On the figure 3B, Scale bar should be presented
Due to our negligence, we forgot to add scale bar when taking pictures, we have already added scale bar.
- Authors should change the resolution of Figure 4D because of it is impossible to analyse the morphology of antler chondrocytes with these microphotographs
We also realize that the pictures are not clear, we zoom in on the picture. In addition, we also implemented cell staining tests.
- Why authors did not use any nuclear living-cell stayning on Figures 5?
Your opinion is very valuable. We have supplemented the experiment of nuclear staining during the revision of the manuscript
- To improve the representation of the experimental results with si-MT1 (Figure 3B), the authors should analyse the colocalisation coefficient between MT1 and DAPI stayning.
DAPI is a nucleic acid dye, and it cannot be described as co-localized with the MT1 protein. DAPI nuclear staining illustrates the localization and expression of MT1 protein, which can better verify the results of Western blot.
- The authors should edit the alphabetical symbols on the figure 5 and legend.
We have explained in the revised manuscript.
- Figure 5 G – pCREB should be normalised with total CREB
We refined the results of this experiment during the revision of the manuscript
2C, 3A, 3C, 3D – should be presented full membrane with marker.
This research has been done for a long time, and the strips of Fig2C and Fig3D did not retain the full film, but we later filled in this part of the results, and we also uploaded supplementary strips. If you think this is not good, we can replace it with a supplementary strip.
- In supplementary original-images of western blot:
I don’t see difference between exposure 8 or 30 sec, even more the membrane after 8 sec represent more intensive bands than 30 sec
This is a mistake we made. In fact, after a single strip is exposed, we just adjusted the light saturation and resolution to reduce the effect of the background. Therefore, we have made corrections in the uploaded WB strips.
Membrane under “Supplementary strips” “exposure 10s” – upper strip – which Ab autors used?
We also detected the p-mTOR protein, but follow-up studies found that the MLT/MT1 signal did not mediate the mTOR signal pathway.
Why in “Original-images pdf file” for western blotting results for figure 4G and 5G the same membrane is represented
Because on that whole membrane we detected AKT protein and Col2a protein at the same time. From the results of the whole membrane, it has been seen that the down-regulation of p-AKT is also accompanied by the down-regulation of Col2a expression.
Reviewer 2 Report
This study has the potential to add an important body of work to the field of antler and bone development. The rationale for addressing the role of melatonin and VEGF in antler growth is clear as are the methods and results. However, there are a few areas that need improving before publishing. Firstly, it would be very helpful to have a native English speaker review and edit the grammar to improve the clarity of the text. In the introduction (P1, line 36), note that antlers grow from pedicles, which are grown once. The annual regrowth of antler is from the pedicles and does not require transformation of stems cells into a pedicle each year. Details of the VEGF-induced degeneration of antler chondrocytes seems to have been omitted from the Methods. It would be helpful to the reader to have a model added of the mechanism via which MLT activates proliferation of mesenchymal cells via Col2a and of inhibiting VEGF-induced degeneration of chondrocytes via Sox9 and PRDX2/4.
Minor edits required:
Figure 1 legend states N=6, while elsewhere the numbers are N=7. Which is correct?
Table 2 has seven periods noted at which antler length was measured. What are the intervals between each time period?
Figure 2 No error bars depicted on 2B, 2C. I understand that the data have been normalised to 1, but there should still be an error term as has been shown for Figure 4E.
It is also unclear what the abbreviation NC is? It seems to be a control – perhaps negative control?
The authors state that VEGF changed the morphology of antler chondrocytes, but it is not clear from Figure 4D what has changed (the figure is small and of low resolution for the review). It would be useful to include some description in the text.
P10, Line325 Write out all abbreviations in full first before abbreviating. Caas =Chinese Academy of Agricultural Sciences (CAAS). Similarly with CST (P11, L377) = Cell Signaling Technology (CST) Biological Reagents Company Ltd, Shanghai, China.
Please also add dilutions at which antibodies were used.
Author Response
Thank you for your comments concerning our manuscript. Your comments are all valuable and very helpful for revising and improving our paper, as well as the important guiding significance to our research. We asked a local English-speaking teacher to help us revise the paper. The modified part has been marked in red font. We agree with your point of view. When the sika deer grows antler for the first time, a pedicle will grow on the forehead crest. After that, antler regeneration will originate from the pedicle every year. In order to be more beneficial to readers, we have drawn a graphical summary (Figure 6).
Figure 1 legend states N=6, while elsewhere the numbers are N=7. Which is correct?
This is a mistake we made while writing, and we have made corrections
Table 2 has seven periods noted at which antler length was measured. What are the intervals between each time period?
We have made an addition (Line,114).
Figure 2 No error bars depicted on 2B, 2C. I understand that the data have been normalised to 1, but there should still be an error term as has been shown for Figure 4E.
Indeed, the data have been normalized to 1. But we made amendments to 2B and 2C in the submitted manuscript.
It is also unclear what the abbreviation NC is? It seems to be a control – perhaps negative control?
NC= negative control, we have made changes (Line130). We have made changes in the revised manuscript.
The authors state that VEGF changed the morphology of antler chondrocytes, but it is not clear from Figure 4D what has changed (the figure is small and of low resolution for the review). It would be useful to include some description in the text.
We also realize that the pictures are not clear, we zoom in on the picture. In addition, we also implemented cell staining tests.
P10, Line325 Write out all abbreviations in full first before abbreviating. Caas =Chinese Academy of Agricultural Sciences (CAAS). Similarly with CST (P11, L377) = Cell Signaling Technology (CST) Biological Reagents Company Ltd, Shanghai, China.
In this article, we did not explain before using abbreviations, which is a bad habit. We will make corrections in future writing. we have made changes (Line 334 and 387).
Please also add dilutions at which antibodies were used.
We have added dilutions in the revised manuscript.